# Assessment of Daily Personal PM2.5 Exposure Level According to Four Major Activities among Children

**Jiyoung Woo [1], Guillaume Rudasingwa [2] and Sungroul Kim [2],***

1    Department of Big Data Engineering, Soonchunhyang University, Asan 31538, Korea; jywoo@sch.ac.kr
2    Department of Environmental Health Sciences, Soonchunhyang University, Asan 31538, Korea; guillaumer1992@gmail.com
*    Correspondence: Sungroul.kim@gmail.com; Tel.: +82-41-530-1266

**Abstract:** Particulate matters less than 2.5 micrometers in diameter (PM2.5), whose concentration has increased in Korea, has a considerable impact on health. From a risk management point of view, there has been interest in understanding the variations in real-time PM2.5 concentrations per activity in different microenvironments. We analyzed personal monitoring data collected from 15 children aged 6 to 11 years engaged in different activities such as commuting in a car, visiting a commercial building, attending an education institute, and resting inside home from October 2018 to March 2019. The fraction of daily mean exposure duration per activity was 72.7 ± 18.7% for resting inside home, 27.2 ± 14.4% for attending an education institute, and 11.5 ± 9.6% and 5.3 ± 5.9% for visiting a commercial building, commuting in a car, respectively. Daily median (interquartile range) PM2.5 exposure amount was 88.9 (55.9–159.7) μg in houses and that in education buildings was 43.3 (22.9–55.6) μg. Real-time PM2.5 exposure levels varied by person and time of day ($p$-value < 0.05). This study demonstrated that our real-time personal monitoring and data analysis methodologies were effective in detecting polluted microenvironments and provided a potential person-specific management strategy to reduce a person's exposure level to PM2.5.

**Keywords:** PM2.5; activity-patterns; real-time; sensor; personal exposure assessment

## 1. Introduction

For decades, particulate matters with aerodynamic diameters less than 2.5 micrometers (PM2.5) have attracted significant attention in the Republic of Korea. According to the Ministry of Environment, the national average concentration of PM2.5 increased from 28 μg/m$^3$ in 2015 to 30 μg/m$^3$ in 2016, and reached 31 μg/m$^3$ in 2017 [1] which is three times higher than the World Health Organization (WHO) annual guideline(10 μg/m$^3$) [2]. Various environmental epidemiology studies have identified significant positive associations between exposure to airborne PM2.5 and the exacerbation of chronic or acute disease symptoms [3,4], increased mortality risk [5], decreased life expectancy [6], and low birth weight [7]. According to the report on the global burden of disease by the WHO, PM2.5 was the ninth major risk factor of disease burden in the Republic of Korea [8] and has been shown to increase the risk of cardiovascular and respiratory diseases, and even premature deaths among Koreans [9]. In general, variation of ambient PM2.5 level is associated with the degree of urbanization, industrialization, and transport as well as ambient chemical reactions under certain meteorological conditions. This implies that a high percentage of people living in Korea are at increased risk of exposure to high levels of PM2.5 as well as adverse health risks as the majority of the population live in urban or industrial areas. Therefore, these days, according to Korean policy makers or other stakeholders, the primary goal of air pollution control policy is to reduce, or ideally eliminate adverse health effects resulting from air pollutants.

The development of an air pollution control policy requires accurate assessment of potential exposure levels according to time and place. The evaluation of the exposure level to PM2.5 within a susceptible population depends critically on how people spend their time at specific sites relating to their activity pattern. Errors in exposure estimation, caused by relying on proxies of approximate exposure duration or level for specific microenvironment, may lead to significant bias in the estimation of related health burdens and developing control policies.

Traditional methods for measuring exposure have sometimes required bulky, heavy, and noisy monitors as well as sophisticated laboratory setups. Alternative approaches relying on sensor-based small portable real-time samplers with activity tracking information can provide a substantial improvement [10] compared with traditional approaches in determining the level of PM2.5 exposure depending on the activities conducted in the specific microenvironments. With recent technology development, a few studies have reported the association of PM2.5 exposure level with time activity patterns in Scotland [11], Portugal [12], India [13], and the United States [14]. Such previous study results indicated that PM2.5 exposure levels were different due to substantial variability of time spent in microenvironments and emphasized personal measurement of near exposure pathways to allow a comprehensive evaluation of the exposure risk which a person might encounter on a daily basis.

In Korea there are still very limited studies that have assessed PM2.5 exposure according to children's activity. Although their daily activity patterns may be different to other countries due to different social conditions or norms i.e., high interest in early education and development resulting in young children attending several types of private education institutes or pre-schools. In this study, we evaluated children's exposure level to PM2.5 by incorporation of PM2.5 data obtained at 10 s intervals and a database of the personal daily activity patterns.

## 2. Materials and Methods

### 2.1. Study Area

We conducted our study in the metropolitan city of Incheon and newly-developed cities of Cheonan and Asan in South Korea. Personal monitoring data were collected during the winter and spring from October 2018 to March 2019. Incheon, located 29 km from Seoul, has a population of 2,920,000. The population of Cheonan and Asan is estimated to be 640,000 and 310,000, respectively. These cities are located on the western side of South Korea, approximately 80 km from Seoul. In general, PM2.5 concentration levels in these areas are high during winter and spring. Moreover, yellow dust also known as Asian dust affects the Korean Peninsula mainly during the spring and winter seasons [15]. Various studies have showed an increased concentrations of PM2.5 during such dust events [15,16], worsening the level of ambient PM2.5.

### 2.2. Study Design and Study Population

We conducted this study with a convenient sampling design. Groups of study participants were recruited from those child patients who had joined our ESCORT (environmental health smart study with connectivity and remote sensing technologies) study [17]. A total of 25 participants aged 6 to 11 years and their parents agreed to carrying personal PM2.5 samplers (please refer to details below) and GPS monitors (Travel Recorder, QSTARZ) and writing their activity and place visited every 30 min on a diary provided (Figure 1). Participants and/or parents recorded the starting time and duration of each specific activity pattern and whether it occurred during a monitoring period.

Over all various activities in their daily activity dairy, we selected 4 major activity pattern types (1) resting inside home, (2) Attending an educational institute, i.e., spending time inside elementary school or kinder garden, (3) spending time inside of car or bus for commuting and (4) spending time inside of other commercial shops including restaurants) which were the most experienced by our study participants. For validation purpose (i.e., outdoor locations), we compared GPS records (one second interval) and daily self-reported activities.

**Figure 1.** An example of the daily activity diary used to collect information of study participant's activity pattern. (Sch: School, KG: Kinder garden, SHS: Secondhand smoke).

This approach offers the opportunity to track spatiotemporal exposure patterns for a group of people sharing common activities in microenvironments from which their exposure profiles spread out in various directions from their residences, education institutes and other possible exposure places (private cars or commercial buildings). A set of a data (PM2.5 concentrations, spatiotemporal GPS information and contextual data) was supposed to be collected by each participant over a period of time, designed to capture everyday activities.

### 2.3. Measurement of PM2.5

The measurements of PM2.5 were conducted using potable real time samplers, MicroPEM (RTI international, Research Triangle Park, NC, USA). We measured PM2.5 concentrations for five days continuously over a 24-h sampling period for each participant. We allowed the placing of monitors in their bedrooms during night time. We used only weekday data because we assumed that children's daily activities were different between weekday and weekend. We finally analyzed data from 15 participants who provided at least 3 weekday activities data.

The MicroPEM allows for integrated sampling with an on-board 780-nm infrared (IR) laser nephelometer operating on a 10.0 s cycling time. In general, MicroPEM is operated for 36 to 40 h with AA alkaline batteries at a 0.5 L/min flow rate. Thus, our field managers asked the parents of a study participant to change the batteries every day using the ones we provided. As a routine calibration procedure, prior to sending our device to participants for data collection, pre-weighed 3.0-μm polytetrafluoroethylene (PTFE) 25-mm TEFLO filters (Zefon International, Ocala, FL, USA) were placed in MicroPEM filter cassettes. MicroPEMs were zeroed with an in-line HEPA filter, and precalibrated at 0.5 L/min with a TSI model 4140 mass flowmeter (TSI, Inc., Shoreview, MN, USA) using Docking Station software (RTI International, Research Triangle Park, NC, USA). Previous study reported that the performance of this portable device was similar to that of a research grade expensive potable monitor, i.e., SIDEPAK [14]. Since our main goal was the evaluation of PM2.5 concentration depending on daily various activities, we used time-series PM2.5 concentration data, and were unable to characterize PM2.5 components [14].

### 2.4. PM2.5 Data Preprocessing

We selected participants whose data were collected at least for three days, and the data collection period was overlapped with other participants. Before we used the 10 s data, we checked the raw data for increasing accuracy and confidence of our data analysis. First, we screened each person's data to

figure out whether or not there was any malfunction of the sampler during the monitoring period; negative values for relative humidity (RH). When such suspected conditions lasted for 5 min, we excluded them from our analysis. If the PM2.5 value was zero but the RH and temperature value were correct, we replaced the zero with their antecedent valid values. Due to person's various activities and different source strengths, our personal data had some outliers. If the PM2.5 value was larger than 1000 μg/m$^3$ in a 10 s interval, we checked the trend of the concentration. If it recorded at one time within one minute, we considered it as a data signal error and excluded it. However, we included the large PM2.5 data if it was an inclining or declining pattern within 5 min. From the raw data with 10 s intervals, if less than 5 missing values were detected in an interval of less than 5 min, we applied the linear regression-based interpolation method and substitute the missing values with expected values obtained from models [10,18,19]. If the missing value interval was longer than 5 min, we excluded them.

### 2.5. Incorporate PM2.5 Data with Activity Pattern Data

Since the time-activity diary investigation was performed on a minute basis, PM2.5 values were combined by matching the values of the closest time of activity pattern after converting 10 s interval data to the mean of PM2.5 with a one-minute interval.

Table 1 shows the number of children engaged in predefined activities in different microenvironments and the respective number of events they performed. Out of 11 predefined activities (7 outdoor, 4 indoor activities), as mentioned earlier, we selected 4 major activities (i.e., commuting in a parents' car; visiting a commercial building other than restaurants; spending time in an education building; spending time at home without cooking) which most participants (at least, n = 12 or larger) experienced. In this study, the event of each major activity was counted if an activity was performed at a different time.

**Table 1.** The number of individuals who experienced the activity predefined and the pooled number of the activity by all participants.

| Activity | No. of Participant | No. of Events |
|:---:|:---:|:---:|
| 1. Commuting with parents' car | 12 | 71 |
| 2. Visiting a commercial building other than restaurant | 14 | 58 |
| 3. Spending time in an education institute (either school, kinder-garden, or private institute) | 12 | 48 |
| 4. Spending time at home without cooking | 15 | 129 |

### 2.6. Measurement of Daily Personal PM2.5 Exposure Amount

Using 10 s real-time PM2.5 concentrations and applying the average daily inhalation rate (0.011 m$^3$/min) for children aged 6 to 11 years, we calculated the daily PM2.5 exposure amount per activity per person according to Equation (1) [20].

Exposure Amount for an activity.

$$(E_j) = \frac{\text{Conc.} \left(\frac{\text{ug}}{\text{m}^3}\right) \times \text{Inh.Rate} \left(0.011 \frac{\text{m}^3}{\text{min}}\right) \times \text{Exp.Duration (sec)}}{60 \text{ sec} / 1 \text{ min}} \tag{1}$$

where Conc. (μg/m$^3$): PM2.5 concentration per activity per person measured every 10 s.
Inh. Rate (0.011 m$^3$/min): Inhalation rate of children aged 6 to 11 years, 99 percentile value
Exp. Duration (second): Exposure duration per activity per person
*j*: Type of activities (*j* = 1 to 11)

Equation (1) indicates the accumulated amount of PM2.5 exposure per activity per child. We set an initial time of 0 and the daily measurement lasted for 24 h. The daily total exposure duration (seconds) should be 24 h × 60 min/h × 60 s/min.

In an ideal case, PM2.5 was recorded every 10 s; however, some time periods had missing values because there was no signal. For the case in which the total measurement data with valid values did not reach to 8640 per day (24 h × 60 min/h × 6 data/min) owing to missing signals, adjustments were made using a weight factor for the exposure duration by dividing the accumulated amount of PM2.5 by the ratio of the actual exposure duration over the total period for each activity. Finally, we obtained the daily total exposure amount (E$_{total}$) by summing up the exposure amount for different activities (Figure 2).

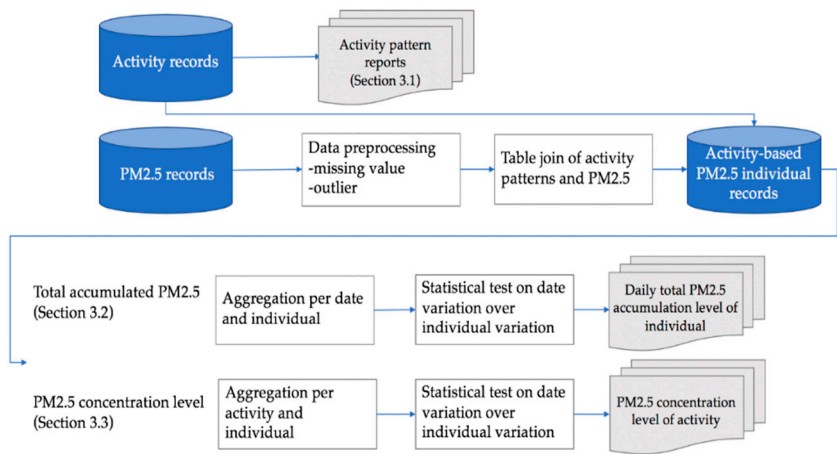

**Figure 2.** Analysis framework for real-time PM2.5 concentration level by activity.

*2.7. Statistical Analysis*

We performed descriptive analysis to obtain the mean standard deviation median and 25 or 75 percentile values. Within a day, the activities with identical names but performed at different times were regarded as unique events with identical event numbers. Daily PM2.5 concentration levels per activity were obtained by dividing the sum of PM2.5 concentration by the number of events for each activity or for each individual (Figure 2).

Since the normality of our PM2.5 exposure data was not guaranteed, we performed the Kruskal–Wallis test to examine whether the daily PM2.5 exposure levels varied among children or activities.

For example, the test statistic for the Kruskal–Wallis test to compare exposure level among 15 children within an activity was

$$H \; = \; (N-1) \frac{\sum_{i=1}^{g} n_i (\overline{r}_i - \overline{r})^2}{\sum_{i=1}^{g} \sum_{j=1}^{n_i} (r_{i,j} - \overline{r})^2} \tag{2}$$

where $r_{i,j}$ is the rank of $E_{total}$ of individual $j$ for activity $i$.
$n_i$ is the number of individuals in group $i$ (i.e., within a single activity ($i$)).
$N$ is the total number of individuals across all groups (days).
$\overline{r}_i$ is the average rank of all individuals in group ($i$).
$\overline{r}$ is the average of all $r_{i,j}$.

Statistical analyses were performed with R (version 3.4.3).

## 3. Results

### 3.1. Daily PM2.5 Exposure Duration

We found that exposure to PM2.5 was dominated by resting inside home (72.7 ± 18.7) and attending an education institute (27.2 ± 14.4) with regard to proportion (%) of daily exposure duration (mean ± standard deviation, SD) per activity; The proportions for visiting a commercial building or commuting in a car were 11.5 ± 9.6% and 5.3 ± 5.9% (Table 2).

**Table 2.** Average time fraction (mean ± SD) that the participants spent in a day (%).

| ID | Car | | Commercial Building | | Education Institute | | Indoor-House | |
|---|---|---|---|---|---|---|---|---|
| | N (Day) | Mean ± SD | N (Day) | Mean ± SD | N (Day) | Mean ± SD | N (Day) | Mean ± SD |
| ID_001 | 2 | 4.4 ± 3.2 | 6 | 11.9 ± 8.6 | 4 | 27.5 ± 2.6 | 7 | 61.5 ± 8.9 |
| ID_002 | 1 | 11.1 ± NA | 3 | 9.2 ± 6.7 | 2 | 38.1 ± 12.5 | 6 | 78.6 ± 31.0 |
| ID_003 | NA | NA ± NA | 1 | 15.5 ± NA | 1 | 77.8 ± NA | 3 | 62.7 ± 36.8 |
| ID_004 | 1 | 1.2 ± NA | 3 | 8.5 ± 4.7 | 4 | 20.5 ± 5.2 | 5 | 73.7 ± 9.5 |
| ID_005 | NA | NA ± NA | 3 | 8.4 ± 0.1 | 3 | 29.1 ± 5.2 | 3 | 56.1 ± 5.5 |
| ID_006 | 2 | 2.7 ± 2.1 | 4 | 3.8 ± 1.5 | 5 | 22.4 ± 6.7 | 7 | 76.7 ± 15.7 |
| ID_007 | 1 | 4.0 ± NA | 1 | 12.7 ± NA | 4 | 18.5 ± 5.8 | 5 | 78.7 ± 6.1 |
| ID_008 | 1 | 0.7 ± NA | 2 | 10.4 ± 4.0 | 4 | 34.6 ± 16.9 | 6 | 70.5 ± 22.4 |
| ID_009 | 1 | 1.9 ± NA | 5 | 13.1 ± 7.7 | 5 | 23.8 ± 1.7 | 5 | 57.7 ± 7.6 |
| ID_010 | NA | NA ± NA | 2 | 10.4 ± 3.0 | NA | NA ± NA | 4 | 94.8 ± 6.2 |
| ID_011 | 7 | 4.2 ± 1.2 | NA | NA ± NA | 8 | 23.6 ± 5.7 | 8 | 71.3 ± 3.2 |
| ID_012 | 6 | 3.3 ± 1.8 | 3 | 9.5 ± 4.8 | 3 | 42.4 ± 35.2 | 7 | 70.9 ± 25.0 |
| ID_013 | 3 | 2.4 ± 0.3 | 1 | 23.9 ± NA | NA | NA ± NA | 3 | 89.7 ± 14.1 |
| ID_014 | 4 | 13.4± 13.7 | 4 | 21.6 ± 25.1 | NA | NA ± NA | 6 | 75.8 ± 24.1 |
| ID_015 | 3 | 9.1 ± 2.5 | 3 | 10.9 ± 6.1 | 2 | 15.6 ± 13.3 | 4 | 77.2 ± 15.7 |
| Total | 14 | 5.3 ± 5.9 | 16 | 11.5 ± 9.6 | 19 | 27.2 ± 14.4 | 19 | 72.7 ± 18.7 |

NA: Not available.

### 3.2. PM2.5 Concentration Level by Activity

As shown in Table 3, there was a variation of PM2.5 concentration according to the activity pattern. In this study, overall median (IQR) PM2.5 concentration was (9.6 (5.7–13.9) $\mu g/m^3$) at an indoor-house and 9.6 (6.6–15.6) $\mu g/m^3$ at an inside commercial building. That inside a parents' car was 10.9 (5.6–18.2) $\mu g/m^3$, and that inside an education building was 11.0 (6.2–14.3) $\mu g/m^3$ (Table 3).

**Table 3.** Percentiles of PM2.5 concentration per person and activity.

| ID | Car | | | | Commercial Building | | | | Education Institute | | | | Indoor-House | | | |
|---|---|---|---|---|---|---|---|---|---|---|---|---|---|---|---|---|
| | N (Event) | 25% | 50% | 75% | N (Event) | 25% | 50% | 75% | N (Event) | 25% | 50% | 75% | N (Event) | 25% | 50% | 75% |
| ID_001 | 5 | 5.7 | 6.3 | 8 | 7 | 5.7 | 7.3 | 12.3 | 4 | 11.7 | 13.9 | 14.7 | 19 | 5.7 | 8.3 | 10.9 |
| ID_002 | 2 | 5.5 | 5.5 | 5.5 | 4 | 2.8 | 4.2 | 9.7 | 3 | 4.1 | 5.5 | 13.9 | 10 | 3.1 | 5.3 | 6.7 |
| ID_003 | NA | _ | _ | _ | 1 | 19.4 | 19.4 | 19.4 | 1 | 28.5 | 28.5 | 28.5 | 3 | 28.3 | 41.5 | 44.1 |
| ID_004 | 1 | 16.4 | 16.4 | 16.4 | 4 | 15.4 | 16.6 | 16.7 | 4 | 14.5 | 14.7 | 17.8 | 9 | 13 | 14.3 | 16.5 |
| ID_005 | NA | _ | _ | _ | 4 | 10.9 | 15.5 | 19.7 | 3 | 4.4 | 4.9 | 5.6 | 9 | 4.7 | 5.8 | 10.2 |
| ID_006 | 4 | 3.9 | 7.8 | 17.5 | 4 | 6.4 | 10.6 | 17.5 | 5 | 9.4 | 14.1 | 21.6 | 15 | 10.1 | 12.2 | 15.3 |
| ID_007 | 2 | 7 | 9.7 | 12.4 | 1 | 60.4 | 60.4 | 60.4 | 4 | 7 | 7.9 | 9 | 7 | 4.6 | 4.7 | 5.1 |
| ID_008 | 1 | 3.2 | 3.2 | 3.2 | 4 | 2.9 | 3 | 3.3 | 6 | 3.1 | 3.5 | 4 | 6 | 5.8 | 9.5 | 11.6 |
| ID_009 | 2 | 20.4 | 24.1 | 27.8 | 11 | 8.6 | 9.8 | 12.3 | 5 | 7.8 | 11 | 20.9 | 14 | 8.7 | 9.8 | 10.9 |
| ID_010 | NA | _ | _ | _ | 2 | 8.6 | 9.5 | 10.4 | NA | _ | _ | _ | 3 | 3.4 | 3.4 | 5.2 |
| ID_011 | 13 | 10.8 | 12.7 | 22.6 | NA | _ | _ | _ | 8 | 10.9 | 11.9 | 12.2 | 9 | 12.2 | 15.5 | 16.7 |
| ID_012 | 15 | 6.6 | 10.9 | 22.1 | 6 | 7.4 | 7.8 | 8.1 | 3 | 25.2 | 36.5 | 49.8 | 9 | 10.8 | 12.6 | 15.4 |
| ID_013 | 7 | 5.9 | 9.6 | 16.7 | 2 | 8.4 | 9.5 | 10.6 | NA | _ | _ | _ | 5 | 4.5 | 7.3 | 16.9 |
| ID_014 | 11 | 9.6 | 17.8 | 26.8 | 5 | 13.7 | 14.5 | 20.5 | NA | _ | _ | _ | 6 | 6.6 | 8.3 | 16.5 |
| ID_015 | 8 | 4.9 | 5.2 | 9 | 3 | 4.4 | 4.9 | 9.3 | 2 | 6.8 | 9 | 11.2 | 5 | 7.8 | 8 | 9.2 |
| Total | 71 | 5.6 | 10.9 | 18.2 | 58 | 6.6 | 9.6 | 15.6 | 48 | 6.2 | 11.0 | 14.3 | 129 | 5.7 | 9.6 | 13.9 |

NA: not available.

*3.3. Daily Exposure Amount of PM2.5 (μg)*

Using Equation (1), we calculated the daily accumulated exposure amount (μg) of PM2.5 per activity and per person (Table 4). Median (IQR) daily exposure amount of PM2.5 were 88.9 (55.9–159.7) μg on staying inside home and 43.3 (22.9–55.6) μg on attending education buildings. In the same way, median (IQR) exposure amounts on spending time in a car or commercial building were 8.0 (3.5–13.2) μg or 13.9 (5.9–23.4) μg. Using the Kruskal–Wallis test, we found that, over the monitoring period, the distributions of daily accumulated exposure amount of PM2.5 per activity or per person were significantly different at the significance level of $p < 0.05$.

**Table 4.** Percentiles of daily accumulated PM2.5 exposure amount (μg) per activity and per person.

| ID | Car | | | | Commercial Building | | | | Education Building | | | | Indoor-House | | | |
|---|---|---|---|---|---|---|---|---|---|---|---|---|---|---|---|---|
| | N (Data) | 25% | 50% | 75% | N (Data) | 25% | 50% | 75% | N (Data) | 25% | 50% | 75% | N (Data) | 25% | 50% | 75% |
| ID_001 | 2 | 4.9 | 8.2 | 11.5 | 6 | 8.3 | 15.4 | 21.4 | 4 | 45.0 | 53.3 | 55.7 | 7 | 54.4 | 84.6 | 135.0 |
| ID_002 | 1 | 9.8 | 9.8 | 9.8 | 3 | 4.0 | 4.2 | 19.6 | 2 | 40.5 | 55.3 | 70.2 | 6 | 58.8 | 63.0 | 78.3 |
| ID_003 | NA | _ | _ | _ | 1 | 20.3 | 20.3 | 20.3 | 1 | 561.7 | 561.7 | 561.7 | 3 | 92.8 | 142.7 | 259.2 |
| ID_004 | 1 | 3.1 | 3.1 | 3.1 | 3 | 17.7 | 29.4 | 29.6 | 4 | 38.9 | 43.0 | 64.2 | 5 | 152.1 | 155.7 | 166.2 |
| ID_005 | NA | _ | _ | _ | 3 | 7.0 | 8.1 | 13.6 | 3 | 20.8 | 21.6 | 23.6 | 3 | 35.9 | 42.3 | 44.9 |
| ID_006 | 2 | 4.5 | 8.1 | 11.8 | 4 | 3.8 | 7.8 | 12.8 | 5 | 23.3 | 38.7 | 81.7 | 7 | 103.7 | 158.9 | 219.0 |
| ID_007 | 1 | 2.9 | 2.9 | 2.9 | 1 | 53.1 | 53.1 | 53.1 | 4 | 21.4 | 24.5 | 30.1 | 5 | 58.6 | 68.6 | 87.0 |
| ID_008 | 1 | 0.4 | 0.4 | 0.4 | 2 | 5.5 | 7.4 | 9.2 | 4 | 12.8 | 18.4 | 32.3 | 6 | 57.5 | 73.5 | 87.8 |
| ID_009 | 1 | 3.6 | 3.6 | 3.6 | 5 | 18.4 | 18.5 | 26.2 | 5 | 30.2 | 42.4 | 68.9 | 5 | 88.6 | 91.2 | 102.0 |
| ID_010 | NA | _ | _ | _ | 2 | 13.3 | 16.3 | 19.2 | NA | _ | _ | _ | 4 | 45.5 | 69.1 | 90.5 |
| ID_011 | 7 | 8.4 | 11.5 | 12.4 | NA | _ | _ | _ | 8 | 46.0 | 47.2 | 48.5 | 8 | 154.3 | 170.7 | 189.0 |
| ID_012 | 6 | 6.5 | 7.5 | 8.5 | 3 | 7.4 | 11.4 | 12.7 | 3 | 101.5 | 148.7 | 189.4 | 7 | 107.8 | 160.5 | 205.6 |
| ID_013 | 3 | 2.7 | 4.4 | 4.6 | 1 | 32.1 | 32.1 | 32.1 | NA | _ | _ | _ | 3 | 87.8 | 143.2 | 158.3 |
| ID_014 | 4 | 15.2 | 31.0 | 48.7 | 4 | 18.4 | 31.9 | 77.6 | NA | _ | _ | _ | 6 | 56.5 | 67.7 | 112.3 |
| ID_015 | 3 | 5.3 | 7.9 | 9.2 | 3 | 6.4 | 9.0 | 9.4 | 2 | 16.7 | 28.8 | 41.0 | 4 | 63.9 | 82.7 | 106.3 |
| Total | 32 | 3.5 | 8.0 | 13.2 | 41 | 5.9 | 13.9 | 23.4 | 45 | 22.9 | 43.3 | 55.6 | 79 | 55.9 | 88.9 | 159.7 |

NA: not available.

## 4. Discussion

This study was conducted to evaluate real-time (10 s interval) personal exposure levels to PM2.5 by four activity patterns among 15 children. We demonstrated that our personal monitoring technologies and incorporation methodologies of various databases were potentially effective to obtain spatial–temporal distributions of PM2.5 per activity and find person-specific possible exposure sources.

In our study, daily median PM2.5 exposure concentrations by activity were similar to each other (commuting with parents' car: 10.9 μg/m$^3$; spending time inside a commercial building: 9.6 μg/m$^3$; spending time inside an education building (elementary school, kinder-garden, nursery etc.): 11.0 μg/m$^3$; and spending time inside home without cooking: 9.6 μg/m$^3$). However, these values were lower than a previous study conducted by Cunha-Lopes et al. (19 μg/m$^3$) [19] or the WHO 24-h exposure guidelines (25 μg/m$^3$). However, we found that depending on the person, the individual daily exposure concentration was close to the acute WHO exposure guidelines.

According to Table 2 and Figure 3, our time-activity pattern records showed that children spent 72.7 ± 18.7% of their time inside home and they inhaled 57.7% of the daily total accumulated amount (dose) of PM2.5. Children spent 27.2 ± 14.4% of their time in education institutes and inhaled 28.1% of the daily total dose. Spending time in commercial buildings or commuting cars accounted for 11.5 ± 9.6 or 5.3 ± 5.9% of the daily total exposure duration and children inhaled 9.0 or 5.2% of their daily total dose there.

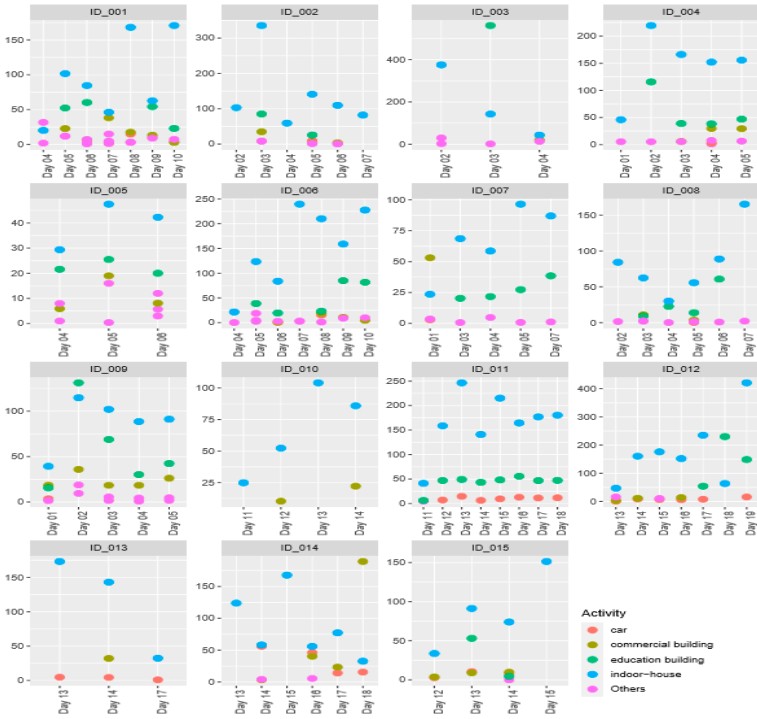

**Figure 3.** Daily accumulated PM2.5 exposure amount per activity and per person.

This study found that children's daily PM2.5 dose at educational institutes (43.3 µg) was approximately half of the amount received at home (88.9 µg) while the ratio of exposure duration at the education institute to that at home was 0.37 ($27.2 \pm 14.4\%$ vs. $72.7 \pm 18.7\%$) indicating that indoor air quality at the education institute or home could determine children's highest exposure level. Thus, application of a management action, i.e., the indoor PM2.5 control strategies for various microenvironments including education institutes as well as homes is immediately necessary to reduce children's exposure to PM2.5 and minimize the risks of potential health effects arising from exposure to PM2.5. On March 2019, the South Korean government passed a revised school health act to improve indoor air quality at pre-schools and primary and secondary schools. By law, the installation of air-cleaning systems and air quality monitoring sensors in classrooms with state government funds is required. A future study on a policy impact assessment is expected.

Lee et al. (2017) have reported the median (interquartile range) of the ratio of $PM_{2.5}$ concentrations with one or two windows open, or with a cooker stove hood operating at concentrations without ventilation: 0.63 (0.40–0.69), 0.41 (0.23–0.56), or 0.17 (0.08–0.25), with dissipation kinetics (($\mu g/m^3$)/min) of 6.5, 20.1, 17.0 or 26.6 respectively, after completion of cooking at an inside Korean apartment [21]. Related to our study results, further studies are recommended to figure out the determinants of indoor air quality at home. Especially, it is highly suggested to conduct a future study to determine the existence of an interactive effect between operating an air purifier and conducting ventilation in a typical classroom and a house in Korea. Also, indoor variation of PM2.5 by children's indoor action (walking or running) at education institutes, and that by infiltration of PM2.5 from outside to inside should be further investigated [22].

A similar outcome has been obtained from personal monitoring with adult study populations. (India [13], Scotland [11] and US [14]) implying that like children, among adults also, as measurements were taken across the heterogeneity of indoor microenvironments, tracking activities influencing personal exposure level to PM2.5 is considered more accurate for exposure data [11].

In this study, we used inexpensive personal monitors i.e., MicroPEM. This monitor was previously evaluated by Sloan et al. (2017) [11]. They reported that performance of MicroPEM was comparable to that of a research-grade expensive portable monitor i.e., SIDEPAK; According to Sloan et al., for

personal samples, 24-h mean PM2.5 concentrations with the MicroPEM were 6.93 μg/m$^3$ (standard error = 0.15) and that for SidePak was 8.47 μg/m$^3$ (standard error = 0.10). In this study, as we mentioned in the method section, we calibrated the MicroPEM before use in the field. The final concentration was provided after adjusting for values of temperature and relative humidity as we did for other inexpensive portable sensor based monitors [23].

In this study, we adopted the continuous monitoring outcomes with a 10 s interval. From the outcomes, we found that data pre-processing was an essential step for maintaining the accuracy of data. Stationary or mobile monitoring data have been often used for estimating exposure level to PM2.5. However, there is a certain limitation of coverage with those stationary monitoring sites. Using a simple but standardized protocol of devices and data mining technologies, more precise and accurate exposure assessment results can be provided to policy makers and the susceptible population [24–26].

## 5. Conclusions and Limitation

This study demonstrated that our methodology of incorporation of PM2.5 data obtained at 10 s intervals and a database of personal daily activity pattern diaries is a useful and feasible method for improving our understanding on the daily variation of exposure levels per person or activity. Our approach introduced in this paper could improve the characterization of exposure patterns and provide a potential person-specific management strategy to reduce a person's exposure level.

In general, a sensor-based study should be interpreted with care. A real-time inexpensive sensor-based approach is convenient to obtain various exposure levels with improved spatial and temporal resolution but it does not guarantee accuracy or precision of the outcome unless a no data quality control procedure is applied. It has been well established that the response of monitoring devices based on light scattering varies with aerosol size distribution, composition, and optical properties need a proper calibration process [27–29]. No single calibration can enable an accurate performance for all particle sources related to diverse activities in microenvironments. Due to our previous comparison test between MicroPEM and Sidepak or PDR-1500 measuring PM2.5 indoors with panfrying and secondhand smoke exposure, we had the experience that MicroPEM was a little noisy but gave comparable results after applying a correction factor. Even though RTI MicroPEM can provide gravimetric measurements but with the daily short sampling duration being limited, applying the gravimetric method for every single site is not practical for routine personal monitoring.

Although we compared self-reported activity patterns with GPS records of longitude and latitude, future advanced tracking technologies with GPS may be helpful and convenient to estimate a person's activity pattern. The three weeks were not mutual for every subject; however, as data collection was conducted from October 2018 to March 2019, those who provided activity data for at least three weekdays were included in the analysis of this study. We originally planned to conduct personal monitoring mutually but owing to the lack of the same monitoring devices, we could not do so. We conducted this study with 15 study participants. Future studies with better sample size need to focus on integration of PM2.5 data with health (respiratory or cardiovascular diseases) databases to provide insights for the reduction of the burden of diseases related to exposure to PM2.5.

As technology continues to improve the capability of inexpensive sensor based personal monitoring, more opportunities will be presented to mine raw data and integrate data into personal health outcomes and current government policies. Personal particulate monitors will become more popular among the susceptible population with underlying susceptible respiratory and cardiovascular conditions who want to monitor particulate levels in order to protect their own health. Integrated personal exposure profiles or combining such profiles, the so called community exposure profile would certainly be valuable to policy makers, as having more detailed data on the polluted microenvironment can motivate policy changes toward PM2.5 reduction. Our study demonstrated that personal-level monitoring with advanced technologies, combined with data analysis skills is very useful to explore and identify local pollution sources.

**Author Contributions:** Conceptualization, S.K.; methodology, S.K. and J.W.; data analysis, J.W.; resources, S.K.; data curation, S.K.; writing—review and editing, S.K., J.W., and G.R.; visualization, J.W. and G.R.; project administration, S.K.; funding acquisition, S.K. All authors have read and agreed to the published version of the manuscript.

**Funding:** This study was funded by the Environmental Health Research Center Project (2016001360003) by the Korea Environmental Industry and Technology Institute, Ministry of Environment, South Korea.

**Acknowledgments:** Authors thank to study participants and their parents.

**Conflicts of Interest:** The authors declare no conflict of interest.

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
