# Peer review of "Assessment of Daily Personal PM2.5 Exposure Level According to Four Major Activities among Children"

_applsci, doi:10.3390/app10010159_

Round 1

Reviewer 1 Report

Dear Authors,

The idea of the paper was very interesting. The experimental setup and the procedure are scientifically sound. However, the quality of the presentation is not appropriate for scientific publications. In particular, 

The quality of the figures needs to be improved. Use accepted fonts such as Times or Calibri for figure fonts as well. Avoid italic fonts. Leave enough space for your figures.  A comprehensive literature review section is necessary.  The conclusion section needs to be rewritten.  Avoid citing under review papers.  The choice of keywords is not suitable.  Numbers need to be spelled out in special occasions see here https://www.scribendi.com/advice/when_to_spell_out_numbers_in_writing.en.html Table numbering is not consistent (see table 2).  The exposure equation used is not clearly defined.  Do not start your paragraphs with conjunctions such as then.  Do not use buzz words such as big data analysis without particularly defining the methods. You can write a separate methodology section about the theoretical methods and statistical analysis that you have used in this project.The statistical methods used need to be clearly explained. 

Since the idea and the experiments conducted are solid, I recommend reconsidering this paper after major revision. I suggest that the authors take advantage of an experienced colleague who has a track record of publications to rewrite the paper. To create a quality publication, you need to allow significant time for your writing. Your grammar and choice of words are acceptable but your technical writing style and structure are poor. 

Author Response

The idea of the paper was very interesting. The experimental setup and the procedure are scientifically sound. However, the quality of the presentation is not appropriate for scientific publications. In particular, 

The quality of the figures needs to be improved. Use accepted fonts such as Times or Calibri for figure fonts as well. Avoid italic fonts. Leave enough space for your figures.  A comprehensive literature review section is necessary. 

>The conclusion section needs to be rewritten. 

Thank you for your comment. We have revised the Conclusion as follows.

This study demonstrated that our real-time personal monitoring and data analysis methodologies are effective in identifying polluted microenvironments and provide a person-specific management strategy to reduce their exposure level.

>Avoid citing under review papers. 

Thank you for your comment. As you recommended, we have changed the references follows.

Previous version: From the raw data with 10 seconds interval, negative (-) PM2.5 value, missing values and outliers were detected and substituted using our own automatic detection and interpolation algorithm published in the separated paper[1]

Revised version: Because of a person’s various activities and sources of strength, our personal data contains some outliers. From the raw data with 10 s interval, negative (-) PM2.5 values, missing values, and outliers were detected and substituted using an automatic detection and interpolation algorithm [24, 25].

Ottosen, T.-B. and P. Kumar, Outlier detection and gap filling methodologies for low-cost air quality measurements. Environmental Science: Processes & Impacts, 2019. 21(4): p. 701-713. Chen, L.-J., et al., ADF: an Anomaly Detection Framework for Large-scale PM2.5 Sensing Systems. IEEE Internet of Things Journal, 2017. PP: p. 1-1.

èThe choice of keywords is not suitable.

Thank you for your comment. We have provided new keywords below.

keywords: PM2.5; Activity-patterns; Real-time; Sensor; Personal exposure assessment;

>Numbers need to be spelled out in special occasions see here https://www.scribendi.com/advice/when_to_spell_out_numbers_in_writing.en.html Table numbering is not consistent (see table 2). 

Thank you for your comment. The numbers in the entire manuscript have been revised.

>The exposure equation used is not clearly defined. 

Thank you for your comment. We formulated Equation (1) to calculate the total accumulated amount of PM2.5 exposure. We provided a detailed explanation of Equation (1) as seen below.

Using 10 s real-time PM2.5 concentrations and applying the average daily inhalation rate (0.011 m3/min) for children aged 6 to 11 years (US EPA, Exposure Factors Handbook), we calculated the daily total accumulated amount of PM2.5 exposure per activity per person according to Equation (1).

Exposure Amount for an activity (Ej)    (Eq.1)

where Conc. (mg/m3): PM2.5 concentration per activity per person measured every 10 s.

         Inh. Rate (0.011 m3/min): inhalation rate of children aged 6 to 11 years,

99 percentile value

      Exp. Duration (seconds): exposure duration per activity per person

       j: type of activities (j = 1 to 11)

Equation 1 indicates the accumulated amount of PM2.5 exposure per activity per child. We set an initial time of 0 and the daily measurement lasted for 24 h. The daily total exposure duration (seconds) should be 24 h, *60 min/h, *60 s/min.

In an ideal case, PM2.5 was recorded every 10 s; however, some time periods had missing values because there was no signal. For the case in which the total measurement data with valid values did not reach to 8,640 per day (24 h, *60 min/h, *6 data/min) owing to missing signals, adjustments were made using a weight factor for the exposure duration by dividing the accumulated amount of PM2.5 by the ratio of the actual exposure duration over the total period for each activity. Finally, we obtained the daily total exposure amount (Etotal) by summing up the exposure amount for different activities. 

>Do not start your paragraphs with conjunctions such as then. 

Thank you for your comment. We have revised the start of the paragraphs.

> Do not use buzz words such as big data analysis without particularly defining the methods.

Thank you for your comment. We changed the term “big data analysis” to “analysis of unstructured data.”

>You can write a separate methodology section about the theoretical methods and statistical analysis that you have used in this project. The statistical methods used need to be clearly explained.

Thank you for your comment. In response to your suggestion, we have separated the theoretical method section and statistical analysis section, and provided details on the statistical analysis methods used.

2.6. Measurement of Daily Personal PM2.5 Exposure Amount

Using 10 s real-time PM2.5 concentrations and applying the average daily inhalation rate (0.011 m3/min) of children aged 6 to 11 years, we calculated the daily total accumulated amount of PM2.5 exposure per activity per person according to Equation (1).

Exposure Amount for an activity (Ej)    (Eq.1)

where Conc. (g/m3): PM2.5 concentration per activity and person measured every 10 seconds.

          Inh. Rate (0.011 m3/min): inhalation rate of children aged 6 to 11 years,

99 percentile value

      Exp. Duration (seconds): exposure duration per activity per person

       j: type of activities (j = 1 to 11)

Equation 1 indicates the accumulated amount of PM2.5 exposure per activity per child. We set an initial time of 0 and the daily measurement lasted for 24 h. The daily total exposure duration (seconds) should be 24 h, *60 min/h, *60 s/min.

In an ideal case, PM2.5 was recorded every 10 s; however, some time periods had missing values because there was no signal. For the case in which the total measurement data with valid values did not reach to 8,640 per day (24 h, *60 min/h, *6 data/min) owing to missing values or outliers (Figure 1), adjustments were made using a weight factor for the exposure duration, in which the accumulated PM2.5 amount was divided by the ratio of actual exposure duration over the total period for each activity. Finally, we obtained the daily total exposure amount (Etotal) by summing up the exposure amount for different activities.

Figure 1. Analysis framework for real-time PM2.5 concentration level by activity

2.7 Statistical analysis.

We performed descriptive analysis to obtain the mean standard deviation median and 25 or 75 percentile values. Within a day, the activities with identical names but performed at different times were regarded as unique events with identical event numbers. Daily PM2.5 concentration levels per activity were obtained by dividing the sum of PM2.5 concentration by the number of events for each activity or for each individual.

Since the normality of our PM2.5 exposure data was not guaranteed, we performed the Kruskal–Wallis test to examine whether the daily PM2.5 exposure levels varied among children or activities. Statistical analyses were performed with R (version 3.4.3).

> Since the idea and the experiments conducted are solid, I recommend reconsidering this paper after major revision. I suggest that the authors take advantage of an experienced colleague who has a track record of publications to rewrite the paper. To create a quality publication, you need to allow significant time for your writing. Your grammar and choice of words are acceptable but your technical writing style and structure are poor. 

Thank you for your comment. We have updated our manuscript accordingly.

Reviewer 2 Report

Dear Editor and authors,

the manuscript titled "Assessment of Daily PM2.5 concentrations according 3 to 4 major activities among children” tries to assess the personal health impact effect from PM2.5 exposure. The experiment follows the state-of-the-art idea to use personal sensors however, I have major concerns about the methodology and the presentation of the whole study. A list of comments that will improve the paper follows:

Lines 9-23: The introduction should give the reader the whole picture of the work that was done in results section. Please rewrite this part. Please also mention the area and the period of the PM monitoring. Lines 21-22: You should highlight the interpretation of this findings. Otherwise this sounds too general. Lines 51-56: The literature review of related works is poor. Please try to update it with resent studies from your area or more widely. Lines 57-60: Please rewrite this part in more details about the scopus of your paper. You suddenly present two predefined groups. Why did you choose these symptom groups? Lines 68-75: This part doesn’t provide any important useful information about the study. Please make this paragraph shorten. Line 77: This map provides useless information. Line 81: Please add a reference about the ESCORT study. Line 112: These three weeks were mutual for all the participants? Lines 118-126: Please add a reference about calibration methodology. Lines 145-150: This part is confusing. Please rewrite and adding an example would be helpful. Lines 152-153: Please rewrite and use more widely used terminology. Line 159: Its not very clear the Scopus of this table. Line169-178: An example would be helpful. This formula was development by you? Any references? Line 179: Please update the whole results part by describing in more details the steps and why you did these calculations. Give details about the Kruskal Wallis test. Which exactly data did you compare? Lines 2019-249: The discussion part is too short Lines 250-278: Limitations of the study should be in a separate section. Line 279: Please try to connect the results with the conclusions and how they respond to the aim of paper (lines 57-60)

Author Response

2nd reviewer

the manuscript titled "Assessment of Daily PM2.5 concentrations according 3 to 4 major activities among children” tries to assess the personal health impact effect from PM2.5 exposure. The experiment follows the state-of-the-art idea to use personal sensors however, I have major concerns about the methodology and the presentation of the whole study. A list of comments that will improve the paper follows:

>Lines 9-23: The introduction should give the reader the whole picture of the work that was done in results section. Please rewrite this part. Please also mention the area and the period of the PM monitoring.

Thank you for your comment. We have updated the Abstract and believe that the current version provides clear picture of our work. As requested, we also included sampling area and period information.

Abstract: PM2.5 has a considerable impact on health whose concentration has increased in Korea. From a risk management point of view, there has been interest in understanding the variations in real-time PM2.5 concentrations per activity in different microenvironments. We analyzed personal monitoring data (five days over 24 h at 10 s intervals) collected from 15 children aged 6 to 11 years engaged in different activities such as  commuting in a car, visiting commercial buildings, attending education institutes, and resting inside homes in Incheon, Cheonan, and Asan from October 2018 to March 2019. The fraction of daily mean exposure duration per activity was 72.7 ± 15.1% for resting inside home, 27.2 ± 14.4 for attending education institutes, and 11.5 ± 9.6, 5.3 ± 5.9 and 3.6 ± 5.6 for visiting commercial buildings, commuting in car, and other activities, respectively. Daily accumulated exposure amount of PM2.5 was 420.9 (321.5 ~ 641.5) mg in house and that in education buildings was 102.2 (47.5 ~ 154.7) mg. Real-time PM2.5 exposure levels varied by person and time of day (p-value < 0.05). This study demonstrated that our real-time personal monitoring and data analysis methodologies are effective in detecting polluted microenvironments and provide a potential person-specific management strategy to reduce a person’s exposure level.

>Lines 21-22: You should highlight the interpretation of this findings. Otherwise this sounds too general.

We appreciate your comment and have provided an interpretation of our findings as seen below.

This study demonstrated that our real-time personal monitoring and data analysis methodologies are effective in detecting polluted microenvironments and provide a potential person-specific management strategy to reduce a person’s exposure level.

>Lines 51-56: The literature review of related works is poor. Please try to update it with resent studies from your area or more widely.

Thank you for your comment. We have updated our literature review and provided more recent information about this study area.

For decades, particulate matter with aerodynamic diameter less than 2.5 µm (PM2.5) has attracted  significant attention in the Republic of Korea. According to the Ministry of Environment, the national average concentration of PM2.5 increased from 28 µg/m³ in 2015 to 30 µg/m³ in 2016, and reached 31 µg/m³ in 2017 [1], which is three times higher than the limit set by the World Health Organization (WHO) (10 μg/m3) [2]. Various environmental epidemiology studies have identified significantly positive associations between exposure to airborne PM2.5 and the exacerbation of chronic or acute disease symptoms [3, 4], increased mortality risk [5], decreased life expectancy [6], and low birth weight [7]. According to the report on global burden of disease by WHO, PM2.5 was the ninth major risk factor of disease in the Republic of Korea [8], and has been shown to increase the risk of cardiovascular and respiratory diseases, and even premature deaths among Koreans [9]. In general, variation of ambient PM2.5 level is associated with the degree of urbanization, industrialization, and transport as well as ambient chemical reactions under certain meteorological conditions. These imply that a high percentage of people living in Korea are at increased risk of exposure to high levels of PM2.5 as well as adverse health risks as the majority of the population live in urban or industrial areas.

Therefore, for Korean policymakers and other stakeholders, the primary goal of air pollution control policy is to reduce or ideally eliminate their adverse effects on human health. The development of an air pollution control policy requires the accurate assessment of potential exposure levels according to time and place. The evaluation of exposure to PM2.5 by a susceptible population can vary depending on how the time they spend at specific locations actually relates to activity patterns. Errors in exposure estimation caused by relying on proxies to approximate exposure duration or levels in a specific microenvironment could lead to significant bias in the estimation of related health burdens and development of control policies.

Traditional methods for measuring exposure have sometimes required bulky, heavy, and noisy monitors as well as sophisticated laboratory setup. Alternative approaches relying on sensor-based small portable real-time samplers with activity tracking information can provide a substantial improvement [16] compared with traditional approaches to determine the level of PM2.5 exposure depending on activities conducted in specific microenvironments such as a restaurant, kitchen, home, store, school, or vehicle.

     Given that there are still very few studies that assess PM2.5 exposure according to children’s activity, in this study, we conducted personal PM2.5 exposure assessments using databases of personal PM2.5 data obtained at 10 s intervals and personal daily activity diaries.

>Lines 57-60: Please rewrite this part in more details about the scopus of your paper. You suddenly present two predefined groups. Why did you choose these symptom groups?

The evaluation of exposure level to PM2.5 within a susceptible population can vary depending on how they spend their time at specific sites and is related to activity patterns. Errors in exposure estimation caused by relying on proxies to approximate exposure duration or levels in a specific microenvironment could lead to significant bias in the estimation of related health burdens and development of control policies.

Traditional methods for measuring exposure have sometimes required bulky, heavy, and noisy monitors as well as sophisticated laboratory setup. Alternative approaches relying on sensor-based small portable real-time samplers with activity tracking information can provide substantial improvement [16] compared with traditional approaches in determining the place of PM2.5 exposure depending on activities conducted in specific microenvironments, such as a restaurant, kitchen, home, store, school, or vehicle.  

     Given that there are still very few studies that assess PM2.5 exposure according to children’s activity, in this study, we conducted personal PM2.5 exposure assessments using incorporated databases of personal PM2.5 data obtained at 10 s intervals and personal daily activity diaries.

>Lines 68-75: This part doesn’t provide any important useful information about the study. Please make this paragraph shorten.

Thank you for your comment. We agree with your observation and have deleted those sentences and shortened the original paragraph as seen below.

We conducted our study in the metropolitan city of Incheon and newly-developed cities of Cheonan and Asan in South Korea. Personal monitoring data were collected during the winter and spring from October 2018 to March 2019. Incheon, located 29 km from Seoul, has a population of 2,920,000. The population of Cheonan and Asan is estimated to be 640,000 and 310,000, respectively. These cities are located in the western side of South Korea, approximately 80 km from Seoul.

In general, PM2.5 concentration levels in these areas are high during winter and spring. Moreover, yellow dust, also known as Asian dust, affects the Korean Peninsula mainly during the spring and winter seasons [16]. Various studies have showed increased concentrations of PM2.5 during such dust events [16, 17], worsening the level of ambient PM2.5.

>Line 77: This map provides useless information.

Response: The map was removed

>Line 81: Please add a reference about the ESCORT study.

Park S, Park C, Lim D, Lee S, Jang S, Yu S, Kim S 

Impact of Indoor Pan-Frying Cooking Activity on Change of Indoor PM2.5 Concentration Level in Asthmatics' Homes Joined to ESCORT study, Journal of Environmental Science International, 2019 28 (10)

>Line 112: These three weeks were mutual for all the participants?

Thank you for your question.

The three weeks were not mutual for every subject; however, as data collection was conducted from October 2018 to March 2019, those who provided activity data for at least three weekdays were included in the analysis of this study. We originally planned to conduct personal monitoring mutually, but owing to the lack of the same monitoring devices, we could not do so. We conducted two group (10 persons per group) monitoring for three weeks (Figure 2).

We explained this in the section on the study's limitation.

>Lines 118-126: Please add a reference about calibration methodology.

 Thank you for your suggestion. This reference has been included.

Ottosen, T.-B. and P. Kumar, Outlier detection and gap filling methodologies for low-cost air quality measurements. Environmental Science: Processes & Impacts, 2019. 21(4): p. 701-713. Chen, L.-J., et al., ADF: an Anomaly Detection Framework for Large-scale PM2.5 Sensing Systems. IEEE Internet of Things Journal, 2017. PP: p. 1-1.

>Lines 152-153: Please rewrite and use more widely used terminology.

Thank you for your suggestion. As requested, we have revised that sentence.

Revised version: Table 1 shows the number of children engaged in predefined activities in different microenvironments and the respective total number of events performed.

Previous version: The number of individuals who experienced the activity predefined and the pooled number of the activity by all participants were presented in Table 1

>Line 159: Its not very clear the Scopus of this table.

Thank you for your comment. We have revised the sentences to clarify the scopus of the table as seen below.

Table 1 shows the number of children engaged in predefined activities in different microenvironments and the respective number of events they performed. Out of 11 predefined activities (7 outdoor, 4 indoor activities), as mentioned earlier, we selected four major activities (i.e., commuting in parents’ car; visiting a commercial building other than restaurants; spending time in an education building; spending time at home without cooking) which most participants (at least, n = 12 or larger) experienced. In this study, the event of each major activity was counted if an activity was performed at different time.

Table 1. The number of individuals who experienced the activity predefined and the pooled number of the activity by all participants.

Activity

No. of participant

No. of events

1. Commuting with parents’ car

12

71

2. Visiting a commercial building other than restaurant

14

58

3. Spending time in an education building (either school, kinder-garden, or private institute)

12

48

4. Spending time at home without cooking

15

129

>Line169-178: An example would be helpful. This formula was development by you? Any references?

Thank you for your comment. We revised the formula to provide clarity. This formula was adopted from the Exposure Factors Handbook published by the US Environmental Protection Agency (EPA/600/R-06/096F). We included corresponding reference. 

>Line 179: Please update the whole results part by describing in more details the steps and why you did these calculations. Give details about the Kruskal Wallis test. Which exactly data did you compare?

Thank you for your question. We provided the details of our data analysis procedure and Kruskal–Wallis test as seen below.

2.6. Measurement of Daily Personal PM2.5 Exposure Amount

Using 10 s real-time PM2.5 concentrations and applying average daily inhalation rate (0.011 m3/min) for children aged 6 to 11 years, we calculated the daily total accumulated amount of PM2.5 exposure per activity per person according to Equation (1) (US EPA, EPA/600/R-06/096F).

Exposure Amount for an activity (Ej)    (Eq.1)

where Conc. (g/m3): PM2.5 concentration per activity and person measured every 10 seconds.

         Inh. Rate (0.011 m3/min): inhalation rate of children aged 6 to 11 years,

99 percentile value

      Exp. Duration (seconds): exposure duration per activity and person

       j: type of activities (j = 1 to 11)

Equation 1 indicates the accumulated PM2.5 exposure amount per activity per child We set an initial time of 0 and the daily measurement lasted for 24 h. The daily total exposure duration (seconds) should be 24 h, *60 min/h, *60 s/min.

In an ideal case, the PM2.5 was recorded every 10 s; however, some time periods had missing values because there was no signal. For the case in which the total measurement data with valid values could not reach to 8,640 per day (24 h, *60 min/h, *6 data/min) owing to missing values or outliers (Figure 1), adjustments were made using a weight factor for the exposure duration by dividing the accumulated PM2.5 amount by the ratio of actual exposure duration over the total period for each activity. Finally, we obtained the daily total exposure amount (Etotal) by summing up the exposure amount for different activities.

Figure 1. Analysis framework for real-time PM2.5 concentration level by activity

2.7 Statistical analysis.

We performed descriptive analysis to provide mean standard deviation median and 25 or 75 percentile values. Within a day, the activities with identical name but performed at different times were regarded as unique events with identical event numbers. Daily PM2.5 concentration levels per activity was obtained by dividing sum of PM2.5 concentration by the number of events for each activity or for each individual.

Since normality of our PM2.5 exposure data was not guaranteed, we performed the Kruskal–Wallis test to examine whether the daily PM2.5 exposure levels varied among 15 children within a day or among daily activities of each person.

For example, the test statistic for the Kruskal–Wallis test to compare exposure levels between 15 children within a day was

where is the rank of  of individual j for date i.

is the number of individuals in group i (i.e., within a single day (date, i))

N is the total number of individuals across all groups (days)

 is the average rank of all individuals in group i

 is the average of all

Statistical analyses were performed with R (version 3.4.3)

>Lines 2019-249: The discussion part is too short

We have added more sentences.

Our time-activity pattern records showed that children spent more than 73% of their time indoors, especially at home, where they received 73% of the daily PM2.5 dose. Spending time in commuting car or commercial building accounted for 5.3 or 11.5 % of the daily time and children inhaled 2.7 or 6.4% of their daily PM2.5 dose. Our work highlights in detecting polluted microenvironments and decreasing exposure duration to reduce children’s PM2.5 inhalation dose, combined with awareness-raising actions for citizens concerning the potential sources.

In this study, we found that children spend approximately a quarter of their time (27.2%) inside education buildings. Our study revealed that the PM2.5 concentration in education buildings (11.0 (6.2~14.3) mg /m3 may have been probably due to students activities (walking or running) in classrooms[4]. The other reason may be effects of infiltration from outside [16, 17]. It has been reported that daily ratio of exposure duration (5.3% = 1.2 hour : 24hour *0.053) in cars was almost half of the duration (11.5%) in commercial building [29] and exposure amount also half. The mean concentration of PM2.5 inside houses or commercial buildings were similar with 10 mg /m3 however the accumulated PM2.5 were way much higher inside. As the PM2.5 accumulate with time, children spent 72.7±18.7 hours indoor house and 27.2± 14.4 hours were spent in education buildings (kindergartens, elementary schools). Our study imply that special care should be applied to improve air quality inside place as well as to reduce exposure duration.

>Lines 250-278: Limitations of the study should be in a separate section.

Thank you for your comment. As requested, we described the limitations of the study in a separate section.

>Line 279: Please try to connect the results with the conclusions and how they respond to the aim of paper (lines 57-60) 

Thank you for your comment. As requested, we have revised our Conclusion as follows.

Children were actively exposed to PM2.5 in various places. For study participants, exposure concentrations varied among the activities of a person as well as among persons for each activity category (p-value < 0.05). This study demonstrated that our real-time personal monitoring and data analysis methodologies are effective in detecting polluted microenvironments and provide a potential person-specific management strategy to reduce a person’s exposure level.

Round 2

Reviewer 1 Report

The conclusion section needs to be rewritten. 

Author Response

Reviewer 1

The conclusion section needs to be rewritten. 

Thank you. As suggested, we updated our conclusion.

Conclusions and Limitations

This study demonstrated that our methodology of incorporation of PM2.5 data obtained at 10 second interval and database of personal daily activity pattern diaries is a useful and feasible method for improving our understanding on daily variation of exposure level per person or activity. Our approach introduced in this paper could improve the characterization of exposure patterns and provide a potential person-specific management strategy to reduce a person’s exposure level. 

In general, sensor based study should be interpreted with care. Real-time inexpensive sensor based approach is convenient to obtain various exposure level with improved spatial and temporal resolution but it does not guarantee accuracy or precision of outcome unless there was no data quality control procedure applied. It is well established that the response of monitoring devices based on light scattering varies with the aerosol size distribution, composition, and optical properties and need a proper calibration process [27-29]. No single calibration can enable accurate performance for all particle sources related to diverse activities in microenvironments. Due to our previous comparison test between MicroPEM and Sidepak or PDR-1500 measuring PM2.5 at indoor with panfrying and secondhand smoke exposure, we had an experience that MicroPEM was a little noisy but comparable results after applying a correction factor. Even though RTI MicroPEM can provide gravimetric measurements but with daily short sampling duration limited applying the gravimetric method for every single site which was not practical for routine personal monitoring.

    Although we compared self-reported activity pattern with GPS record of longitude and latitude, future advanced tracking technologies with GPS may be helpful and convenient to estimate person’s activity pattern. The three weeks were not mutual for every subject; however, as data collection was conducted from October 2018 to March 2019, those who provided activity data for at least three weekdays were included in the analysis of this study. We originally planned to conduct personal monitoring mutually, but owing to the lack of the same monitoring devices, we could not do so. We conducted this study with 15 study participants. Future studies with better sample size need to focus on integration with PM2.5 data with health (respiratory or cardiovascular diseases) databases to provide insights for reduction of burden of disease with related to exposure to PM2.5.

As technology continues to improve the capability of inexpensive sensor based personal monitoring, more opportunities will be faced to mining raw data and integrate data into personal health outcome and current government policies. Personal particulate monitors will become more popular among susceptible population with underlying susceptible respiratory and cardiovascular conditions who want to monitor particulate levels in order to protect their own health. Integrated personal exposure profiles or combined such profiles, so called community exposure profile would certainly be valuable to policy makers, as having more detailed data on polluted microenvironment which can motivate policy changes toward PM2.5 reduction. Our study demonstrated that personal-level monitoring with advanced technologies with data analysis skill is very useful to explore and identify local pollution source.

Reviewer 2 Report

Dear authors,

Thank you for the updated manuscript. It has been improved, however some parts still need to be clarified. Also, extensive English editing is needed.

The introduction part has been partially updated. (similar studies are not mentioned) In the discussion part only comparison with Cunha-Lopes et al. 2019 was done. PM sensors calibration should be mentioned in chapter “3 Measurement of PM2.5” The limitations of the study should be in a separate part and not in the discussion section-discussion section is limited. Lines 135-136. The way you managed missing data should be described in more details. Lines 144-146. Still confusing. Conclusions part is too short.

Author Response

Reviewer 2

The introduction part has been partially updated. (similar studies are not mentioned) (GUI)

Thank you. As seen below, we included other similar studies report recently and we updated the introduction.

With recent technology development, a few study have reported the association of PM2.5 exposure level with time activity patterns in Scotland [11], Portugal [12], India [13] and United States [14]. Such previous study results indicated that PM2.5 exposure levels were different due to substantial variability of time spent in microenvironments and emphasized personal measurement of near exposure pathways to allow a comprehensive evaluation of the exposure risk which a person might encounter on a daily basis.

In Korea there are still very limited studies that assessed PM2.5 exposure according to children’s activity. Although their daily activity patterns may be different to other countries due to different social conditions or norms i.e., high interest in early education and development resulting in young children’s attending to several types of private education institutes or preschools. In this study, we evaluated children’s exposure level to PM2.5 by incorporation of PM2.5 data obtained at 10 second interval and database of personal daily activity patterns.

In the discussion part only comparison with Cunha-Lopes et al. 2019 was done.

Thank you. We included recent similar studies conducted

In our study, daily median PM2.5 exposure concentrations by activity were similar to each other (commuting with parents’ car: 10.9 µg/m3; spending time inside commercial building: 9.6 µg/m3; spending time inside education building (elementary school, kinder-garden, nursery etc…): 11.0 µg/m3; and spending time inside home without cooking: 9.6 µg/m3). But these values were lower than a previous study conducted by Cunha-Lopes et al. (19 µg/m3) [19] or the WHO 24-hour exposure guidelines (25 µg/m3). However, we found that depending on person, individual daily exposure concentration was close to the acute WHO exposure guidelines.

Our time-activity pattern records showed that children spent 72.7 ± 18.7 % of their time at inside home and they inhaled 57.7 % of the daily total accumulated amount (dose) of PM2.5. Children spent the 27.2 ± 14.4 % of their time in education institutes and inhaled 28.1 % of the daily total dose. Spending time in a commercial buildings or a commuting cars accounted for 11.5 ± 9.6 or 5.3 ± 5.9 % of the daily total exposure duration and children inhaled 9.0 or 5.2 % of their daily total dose there.

This study found that children’s daily PM2.5 dose at educational institutes (43.3 mg) was approximately a half of the amount received at home (88.9 mg) while the ratio of exposure duration at education institute to that at home was 0.37 (27.2 ± 14.4 % vs. 72.7 ± 18.7 %) indicating that indoor air quality at education institute or home could determine children’ most exposure level. Thus, application of a management action, i.e., the indoor PM2.5 control strategies for various microenvironment including education institute as well as home is immediately necessary to reduce children’s exposure to PM2.5 and minimize the risks of potential health effects arising from exposure to PM2.5. On March 2019, The South Korean government passed a revised school health act to improve indoor air quality at preschools and primary and secondary schools. By the law, it has been required the installation of air-cleaning systems and air quality monitoring sensors in classrooms with state government fund. A future study of policy impact assessment is expected.

Lee et al. (2017) have reported the median (interquartile range) of the ratio of PM2.5 concentrations with one or two windows open, or with cooker stove hood operating to the concentrations without ventilation: 0.63 (0.40–0.69), 0.41 (0.23–0.56), or 0.17 (0.08–0.25), with dissipation Kinetics ((μg/m3)/min) of 6.5, 20.1, 17.0 or 26.6 respectively, after completion of cooking at inside Korean apartment [21]. Related to our study results, further studies are recommended to figure out determinants of indoor air quality at home. Especially, it is highly suggested conducting a future study to figure out existence of an interactive effect between operating an air purifier and conducting ventilation in a typical classroom and house of Korea. Also, indoor variation of PM2.5 by children’s indoor action (walking or running) at education institutes, and that by infiltration of PM2.5 from outside to inside should be further investigated [22].

A similar outcome has been obtained from personal monitoring with adult study population. (India [13], Scotland[11] and US [14]) implying that like children, among adults also, as measurements were taken across the heterogeneity of indoor microenvironments, tracking activities influencing personal exposure level to PM2.5 is considered for more accurate exposure data [11].

In this study, we used inexpensive personal monitors i.e., MicroPEM. This monitor was previously evaluated by Sloan et al (2017) [11]. They reported that performance of MicroPEM was comparable to that of research-grade expensive portable monitor i.e., SIDEPAK; According to Sloan et al., for personal samples, 24-hr mean PM2.5 concentrations with the MicroPEM were 6.93 μg/m3 (standard error = 0.15) and that for SidePak was 8.47 μg/m3 (standard error = 0.10). In this study, as we mentioned in the method section, we calibrated the MicroPEM before we used in the field. And, final concentration was provided after adjusting for values of temperature and relative humidity as we done for other inexpensive portable sensor based monitors [23].

Steinle, S.; Reis, S.; Sabel, C.E.; Semple, S.; Twigg, M.M.; Braban, C.F.; Leeson, S.R.; Heal, M.R.; Harrison, D.; Lin, C., et al. Personal exposure monitoring of PM2.5 in indoor and outdoor microenvironments. Sci Total Environ 2015, 508, 383-394, doi:10.1016/j.scitotenv.2014.12.003. Cunha-Lopes, I.; Martins, V.; Faria, T.; Correia, C.; Almeida, S.M. Children's exposure to sized-fractioned particulate matter and black carbon in an urban environment. Building and Environment 2019, 155, 187-194, doi:https://doi.org/10.1016/j.buildenv.2019.03.045. Pant, P.; Habib, G.; Marshall, J.D.; Peltier, R.E. PM2.5 exposure in highly polluted cities: A case study from New Delhi, India. Environ Res 2017, 156, 167-174, doi:10.1016/j.envres.2017.03.024. Sloan, C.D.; Philipp, T.J.; Bradshaw, R.K.; Chronister, S.; Barber, W.B.; Johnston, J.D. Applications of GPS-tracked personal and fixed-location PM2.5 continuous exposure monitoring. Journal of the Air & Waste Management Association 2016, 66, 53-65, doi:10.1080/10962247.2015.1108942. Lee , S., Yu, S., Kim, S., Evaluation of Potential Average Daily Dose (ADDs) of PM2.5 for homemakers conducting Pan-Frying Inside Ordinary Homes under Four Ventilation conditions. International Journal of Environmental Research and Public Health, 2017. 14(1): 78. Peng, Z.; Deng, W.; Tenorio, R. Investigation of Indoor Air Quality and the Identification of Influential Factors at Primary Schools in the North of China. Sustainability (Switzerland) 2017, 9, doi:10.3390/su9071180. Kim, S.; Park, S.; Lee, J. Evaluation of Performance of Inexpensive Laser Based PM2.5 Sensor Monitors for Typical Indoor and Outdoor Hotspots of South Korea. Applied Sciences 2019, 9, 1947, doi:10.3390/app9091947.

PM sensors calibration should be mentioned in chapter “3 Measurement of PM2.5” (GUI)

Thank you. As asked, we mentioned calibration procedure in the section of “2.3 Measurement of PM2.5.” as seen below.

The MicroPEM allows for integrated sampling with an on-board 780-nm infrared (IR) laser nephelometer operating on a 10.0 seconds cycling time. In general, MicroPEMs are operated for 36 to 40 hours with AA alkaline batteries at a 0.5 L/min flow rate. Thus, our field managers asked parents of a study participant changing battery every day using the ones we provided. As a routine calibration procedure, prior to sending our device to participants for data collection, pre-weighed 3.0-µm polytetrafluoroethylene (PTFE) 25-mm TEFLO filters (Zefon International, Ocala, FL) were placed in MicroPEM filter cassettes. MicroPEMs were zeroed with an in-line HEPA filter, and precalibrated at 0.5 L/min with a TSI model 4140 mass flowmeter (TSI, Inc., Shoreview, MN) using Docking Station software (RTI International, Research Triangle Park, NC). According to previous study, the performance of this inexpensive portable device was similar to research grade expensive potable monitor, i.e., SIDEPAK (Sloan et al., Kim et al.). Since our main goal was evaluation of PM2.5 concentration depending on daily various activities, we used time-series PM2.5 concentration data, and were unable to characterize PM2.5 components[14].

Sloan, C.D., et al., Applications of GPS-tracked personal and fixed-location PM2.5 continuous exposure monitoring. Journal of the Air & Waste Management Association, 2016. 66(1): p. 53-65.

The limitations of the study should be in a separate part and not in the discussion section-discussion section is limited.

Thank you. We provided the limitation of the study in a separated section.

Conclusions and Limitations

This study demonstrated that our methodology of incorporation of PM2.5 data obtained at 10 second interval and database of personal daily activity pattern diaries is a useful and feasible method for improving our understanding on daily variation of exposure level per person or activity. Our approach introduced in this paper could improve the characterization of exposure patterns and provide a potential person-specific management strategy to reduce a person’s exposure level. 

In general, sensor based study should be interpreted with care. Real-time inexpensive sensor based approach is convenient to obtain various exposure level with improved spatial and temporal resolution but it does not guarantee accuracy or precision of outcome unless there was no data quality control procedure applied. It is well established that the response of monitoring devices based on light scattering varies with the aerosol size distribution, composition, and optical properties and need a proper calibration process [27-29]. No single calibration can enable accurate performance for all particle sources related to diverse activities in microenvironments. Due to our previous comparison test between MicroPEM and Sidepak or PDR-1500 measuring PM2.5 at indoor with panfrying and secondhand smoke exposure, we had an experience that MicroPEM was a little noisy but comparable results after applying a correction factor. Even though RTI MicroPEM can provide gravimetric measurements but with daily short sampling duration limited applying the gravimetric method for every single site which was not practical for routine personal monitoring.

    Although we compared self-reported activity pattern with GPS record of longitude and latitude, future advanced tracking technologies with GPS may be helpful and convenient to estimate person’s activity pattern. The three weeks were not mutual for every subject; however, as data collection was conducted from October 2018 to March 2019, those who provided activity data for at least three weekdays were included in the analysis of this study. We originally planned to conduct personal monitoring mutually, but owing to the lack of the same monitoring devices, we could not do so. We conducted this study with 15 study participants. Future studies with better sample size need to focus on integration with PM2.5 data with health (respiratory or cardiovascular diseases) databases to provide insights for reduction of burden of disease with related to exposure to PM2.5.

As technology continues to improve the capability of inexpensive sensor based personal monitoring, more opportunities will be faced to mining raw data and integrate data into personal health outcome and current government policies. Personal particulate monitors will become more popular among susceptible population with underlying susceptible respiratory and cardiovascular conditions who want to monitor particulate levels in order to protect their own health. Integrated personal exposure profiles or combined such profiles, so called community exposure profile would certainly be valuable to policy makers, as having more detailed data on polluted microenvironment which can motivate policy changes toward PM2.5 reduction. Our study demonstrated that personal-level monitoring with advanced technologies with data analysis skill is very useful to explore and identify local pollution source.

Lines 135-136. The way you managed missing data should be described in more details. (JIYOUNG)

Thank you. As asked we provided additional description about the way we managed missing values as seen below.

We selected participants whose data were collected at least for three days, and the data collection period was overlapped with other participants. Before we use the ten second data, we checked the raw data for increasing accuracy and confidence of our data analysis. First, we screened each person’s data to figure out whether or not there was any malfunction of sampler during monitoring period; negative values for relative humidity (RH). When such suspected conditions lasted for a five minutes, we excluded them from our analysis. If PM2.5 value was zero but RH and temperature value were correct, we replaced the zero with their antecedent valid values. Due to person’s various activities and different source strengths, our personal data has some outliers. If PM2.5 value were larger than 1000 mg/m3 in ten seconds interval, we checked the trend of the concentration. If it recorded at one time within one minute, we considered it as a data signal error and excluded it. However, we included those large PM2.5 data if it was an inclining or declining pattern within five minutes. From the raw data with ten seconds interval, if less than 5 missing values were detected in an interval of less than five minutes, we applied the linear regression-based interpolation method and substitute the missing values with expected values obtained from models [10, 18, 19]. If missing value interval was longer than five minutes, we excluded them.

Lines 144-146. Still confusing.

Thank you. We agree with you. To provided clarity, we deleted the sentences which were not necessary.

Since time-activity diary investigation was performed on minute basis, PM2.5 values were combined by matching the values of the closest time of activity pattern after converting ten seconds interval data to mean of PM2.5 with one-minute interval.

Table 1 shows the number of children engaged in predefined activities in different microenvironments and the respective number of events they performed. Out of 11 predefined activities (7 outdoor, 4 indoor activities), as mentioned earlier, we selected four major activities (i.e., commuting in parents’ car; visiting a commercial building other than restaurants; spending time in an education building; spending time at home without cooking) which most participants (at least, n = 12 or larger) experienced. In this study, the event of each major activity was counted if an activity was performed at different time.

Table 1. The number of individuals who experienced the activity predefined and the pooled number of the activity by all participants.

Activity

No. of participant

No. of events

1. Commuting with parents’ car

12

71

2. Visiting a commercial building other than restaurant

14

58

3. Spending time in an education institute

(either school, kinder-garden, or private institute)

12

48

4. Spending time at home without cooking

15

129

Before conducting data analysis, we compared inter-intra correlation coefficient to compare whether or not variation of PM2.5 concentrations by activity patterns within each person over five days was larger than that of PM2.5 between participants. Since it turned out that the variation within person were larger, we reported person’s mean PM2.5 concentrations by daily activity over five-day real time data. Then, we obtained overall mean PM2.5 concentration value by each person’s four typical activity patterns.

Conclusions part is too short.

Thank you. We updated our conclusion.

This study demonstrated that our methodology of incorporation of PM2.5 data obtained at 10 second interval and database of personal daily activity pattern diaries is a useful and feasible method for improving our understanding on daily variation of exposure level per person or activity. Our approach introduced in this paper could improve the characterization of exposure patterns and provide a potential person-specific management strategy to reduce a person’s exposure level. 

In general, sensor based study should be interpreted with care. Real-time inexpensive sensor based approach is convenient to obtain various exposure level with improved spatial and temporal resolution but it does not guarantee accuracy or precision of outcome unless there was no data quality control procedure applied. It is well established that the response of monitoring devices based on light scattering varies with the aerosol size distribution, composition, and optical properties and need a proper calibration process [27-29]. No single calibration can enable accurate performance for all particle sources related to diverse activities in microenvironments. Due to our previous comparison test between MicroPEM and Sidepak or PDR-1500 measuring PM2.5 at indoor with panfrying and secondhand smoke exposure, we had an experience that MicroPEM was a little noisy but comparable results after applying a correction factor. Even though RTI MicroPEM can provide gravimetric measurements but with daily short sampling duration limited applying the gravimetric method for every single site which was not practical for routine personal monitoring.

    Although we compared self-reported activity pattern with GPS record of longitude and latitude, future advanced tracking technologies with GPS may be helpful and convenient to estimate person’s activity pattern. The three weeks were not mutual for every subject; however, as data collection was conducted from October 2018 to March 2019, those who provided activity data for at least three weekdays were included in the analysis of this study. We originally planned to conduct personal monitoring mutually, but owing to the lack of the same monitoring devices, we could not do so. We conducted this study with 15 study participants. Future studies with better sample size need to focus on integration with PM2.5 data with health (respiratory or cardiovascular diseases) databases to provide insights for reduction of burden of disease with related to exposure to PM2.5.

As technology continues to improve the capability of inexpensive sensor based personal monitoring, more opportunities will be faced to mining raw data and integrate data into personal health outcome and current government policies. Personal particulate monitors will become more popular among susceptible population with underlying susceptible respiratory and cardiovascular conditions who want to monitor particulate levels in order to protect their own health. Integrated personal exposure profiles or combined such profiles, so called community exposure profile would certainly be valuable to policy makers, as having more detailed data on polluted microenvironment which can motivate policy changes toward PM2.5 reduction. Our study demonstrated that personal-level monitoring with advanced technologies with data analysis skill is very useful to explore and identify local pollution source.
